# Inositol Hexaphosphate in Bone Health and Disease

**DOI:** 10.3390/biom14091072

**Published:** 2024-08-27

**Authors:** Yuji Yoshiko, Ivana Vucenik

**Affiliations:** 1Pi Skovy, 1-15-31-9, Mukainadahonmachi, Minami-ku, Hiroshima 734-0062, Japan; 2Department of Medical and Research Technology, University of Maryland School of Medicine, Baltimore, MD 21201, USA; ivucenik@som.umaryland.edu

**Keywords:** myo-inositol hexaphosphate, bone formation, bone resorption, laboratory animals, humans

## Abstract

Dietary phytic acid/phytate/myo-inositol hexaphosphate (IP6), a phosphate reservoir in plants, was viewed as antinutrient, caused by an influence on the bioavailability of minerals through its chelating activity. However, there is a growing body of evidence indicating that IP6 has beneficial (e.g., antiinflammatory, antibacterial, and anticancer) effects on multiple biological processes. Also, IP6 and its metabolites are known to exist in mammalian cells, including human cells, and the role of IP6 as a functional molecule is attracting attention. IP6 can bind to the growth sites of hydroxy-apatite (HA) and calcium oxalate crystals to prevent their growth and hence inhibit pathological calcification. SNF472, hexasodium IP6, is currently being evaluated in clinical studies as a treatment for vascular calcification and calciphylaxis. However, since HA crystal growth within bone matrix is an essential process in bone formation, it is possible that IP6 intake may inhibit physiological mineralization and bone formation, although currently more published studies suggest that IP6 may contribute to bone health rather than inhibit bone formation. Given that IP6 and its metabolites are thought to have diverse activities and many health benefits, it remains important to consider the range of effects of IP6 on bone.

## 1. Introduction

Phytic acid/phytate/myo-inositol hexaphosphate (hereinafter referred to as IP6) is found in high amounts (e.g., at levels of 0.5% to 3% of dry weight in major food sources of IP6) and generally as an insoluble calcium/magnesium salt (phytin) in many edible seeds, legumes, nuts, and whole grains, where it acts as a major plant phosphate store. Inositol phosphates (IPs) including IP6 are known to be present in mammalian cells, where their functions are under investigation [1,2,3,4]. In 1934, Bruce and Callow reported that the rachitic pathology induced in 6-week-old rats fed with a routine rachitogenic diet were alleviated by supplementation with 14 g/kg IP6 sodium salt (does not exist in nature) but not IP6 calcium/magnesium salt (9 g/kg) for 14 days [5]. The authors also reported that these results were obtained only in rats fed a high-calcium/low-phosphorus diet and that IP6 had anticalcification activity by forming an insoluble calcium complex and reducing calcium bioavailability. However, as of 1951, Walker argued that the significance of IP6 as an anticalcification factor was not found in various nutritional environments in humans [6].

There are many experimental animal models in which IP6 has been evaluated as a source of nutrients and antinutrients and as a bioactive molecule. Given that the effects of sodium IP6 and calcium/magnesium IP6 are not equivalent [5], it should be noted that many studies have used non-naturally occurring IP6 sodium salts. It is also true that studies using sodium IP6 have not yielded consistent results. For example, SD rats were fed the AIN-93G diet with or without sodium IP6 (1, 3, or 5%) for 12 weeks after weaning [7]. In this model, rats fed with high doses of sodium IP6 (3 and 5%) exhibited decreases in growth rate, fat-free lean mass, whole-body fat, mineral absorption (calcium, magnesium, and iron), hypophosphatemia, hypomagnesemia, hypocalcemia (only in 5%), marked increases in parathyroid hormone (PTH) levels and the BUN/creatinine ratio, kidney injury including calcium deposits, and bone loss. The lack of antinutrient effects of IP6 on levels of calcium, magnesium, zinc, iron, and copper in test samples including bone (femur) and plasma was reported in 21-day-old female Wistar rats fed the AIN-76A diet with or without 1% sodium IP6 for 12 weeks [8]. The latter authors showed that IP6 suppressed only iron levels in the brain and suggested the consumption of foods containing high levels of organic acids, such as ascorbic acid (AA), to counteract this inhibitory effect of sodium IP6.

In monogastric livestock that do not have phytase, an IP6-degrading enzyme, and depend primarily on grains as a source of nutrients, little bioavailability of phosphorus in calcium/magnesium IP6 can be expected. In addition, the formation of insoluble complexes between IP6 and minerals leads to decreased bioavailability of nutrients including minerals. These events result in increased feed intake with higher costs and the excretion of undigested nutrients contributes to environmental pollution [9,10]. In children living in the majority of developing countries, low dietary calcium intake when an IP6-rich diet is consumed and vitamin D deficiency have been identified as issues [11]. However, consistent with Walker’s report [6], recent real-world data in healthy humans have yet to demonstrate mineral deficiencies induced by dietary IP6 intake [12]. It is a matter of fact that IP6 is widely used as an excipient in processed foods, cosmetics, and pharmaceutical preparations, and its safety profile is excellent. Further, a number of studies have evaluated the beneficial effects of IP6 on human health, including its antidiabetic, antiinflammatory, antioxidant, anticancer, and lipid-lowering activities [1,2,3,13,14,15]. In any case, nutritional imbalance and/or overdose should be treated with caution in some cases. For example, an excess of vitamin D in blood can cause nausea, vomiting, muscle weakness, excessive urination, renal dysfunction, and so on (https://ods.od.nih.gov/factsheets/list-all/, accessed on 10 January 2024; see also [16]). Using juvenile Lister Hooded rats fed with various diets including vitamin D, calcium, cereals, and sodium IP6, Roberts and Yudkin reported that IP6 in ordinary diets had varying effects on bone calcification (ash content), depending on other components in the diet [17]. Thus, it is important to continue to study the health risk of excess intake of IP6, in association with the balance of nutrients including minerals and vitamin D, the biosynthesis of active vitamin D_3_, and the functions of organs/tissues involved in mineral metabolism.

As our understanding of the biosynthesis of various Ips, including IP6, has increased [1,2,3,4], so too has the need to understand their potential functions in mammalian cells increased [18]. The biodistribution (brain, kidney, bone, plasma, and urine) of IP6 was found to depend on dietary intake of IP6 in postweaning female Wistar rats fed the AIN-76A diet with or without 1% sodium IP6, the AIN-76A diet with or without 10 g/kg sodium IP6, and the standard non-purified diet UAR A03 [19,20]. From these studies, the authors concluded that the majority of IP6 in organs and tissues has a dietary origin. In addition, subjects with higher IP6 intake on the Mediterranean diet had higher urinary IP6 concentrations than those with lower intake, suggesting that some IP6 is also absorbed from the diet in humans [21,22,23]. Such findings have increased interest in understanding the biological actions of dietary IP6 in mammals.

Mineralization of bones and teeth proceeds by crystal growth of calcium phosphate bound to organic extracellular matrix (ECM), which comprises mainly collagen fibers with lower amounts of multiple non-collagenous proteins and proteoglycans. Bone-forming osteoblasts synthesize and deposit unmineralized ECM (osteoid), followed by release of extracellular vesicles (so-called matrix vesicles) into the osteoid. In matrix vesicles, calcium and phosphate accumulate via the activity of calcium channels and phosphate transporters, respectively, and form hydroxyapatite (HA) crystals, which then grow extracellularly, resulting in completion of calcification [24]. Dentin-forming odontoblasts and cementum-forming cementoblasts are thought to have similar systems, while the embryologically distinct enamel-forming ameloblasts and enamel calcification are distinct. Ninety-nine percent and 85% of calcium and phosphate, respectively, are localized in bone, and this large mineral pool is regulated by bioactive molecules including hormones such as vitamin D_3_, PTH, and fibroblast growth factor 23 [25]. Disruption of these regulatory mechanisms due to aging and diseases causes bone loss with pathological calcification in various tissues including the vasculature [26]. The majority of ectopic calcification comprises calcium phosphate (HA), as seen normally in bones and teeth, while calcium oxalate also contributes to urinary stones. IP6 and its hydrolysates inhibit growth of these crystals in vitro and ameliorates pathological calcification in multiple experimental animal models and humans [27]. Based on such activity, SNF472 (hexasodium IP6) is undergoing clinical trials in the United States as a vascular calcification inhibitor [28,29,30,31]. A major mechanism underlying the anticalcification activity of IP6 in pathological situations is thought to be the binding of IP6 to growing HA crystals through negatively charged phosphate groups [32]. If this also occurs under physiological conditions, IP6 may impinge on normal bone formation. We therefore summarize information on the effects of IP6 on mammalian bone with focus on advances during the last 50 years and highlight current issues and future prospects. While the effects of IP6 on mineral availability and metabolism in humans and animals including chickens and pigs have long been reported, we focus here on studies in human and laboratory animal bones. 

## 2. The Effects of IP6 on Bone Cells (Osteoblasts and Osteoclasts) In Vitro

The first article on the effect of IP6 on calcification of hard tissues appeared in 1972 and reported inhibition of mineralization of bovine collagen sections and tibial sections from rachictic rats by partly hydrolyzed preparations of IP6; of note, neither less than 25% nor more than 55% hydrolyzed IP6 prevented mineralization in these models [33]. Sodium IP6 was found to inhibit mineralization in human enamel samples [34], but less consistent results were found in other studies [35]. In recent studies on humans (18–85 years of age) using fluoride toothpaste, sodium IP6 was found to have no effect on the remineralizing ability of fluoride in model caries enamel specimens placed into a partial denture [36]. As for osteoblastic cells, Brighton et al. reported in 1992 that IP6 and IP metabolites were involved in DNA synthesis in newborn rat calvaria cells under cyclical biaxial mechanical strain conditions [37]. IP6 levels in ATDC5, a mouse chondrogenic cell line, decreased in association with the expression of MINPP1, an enzyme hydrolyzing IP6 and IP5 and involved in chondrogenic differentiation [38]. Although these findings suggest the importance of intracellular IP6 in osteo/chondrogenic cells, Minpp1-deficient mice were found to be viable and exhibited apparently normal bone and cartilage development [39]. Deletion of MINPP1, an endoplasmic reticulum-resident enzyme, in embryonic fibroblasts was shown to increase cytoplasmic IP5 and IP6 levels and upregulate a novel substitutive inositol polyphosphate phosphatase, suggesting the importance of multiple inositol polyphosphate phosphatases in skeletal cell function. Recently, an early-onset neurodegenerative syndrome was described in loss-of-function mutations of MINPP1 [40]. Further studies are therefore needed to determine the roles of intracellular IP6, together with other IPs and MINPP1, in osteo/chondrogenic cells. 

Table 1 summarizes studies on the effects of IP6 on bone cells in vitro. The first evaluation of the effects of exogenously supplied IP6 appears to be the study of Agarwal et al. using the mouse osteoblastic cell line MC3T3 [41]. They showed that sodium IP6 and other IPs such as sodium IP4 and sodium IP5 promoted apoptosis (maximum concentrations of 100 µM) under serum-starved culture conditions and that treatment with sodium fluoride, which increases levels of endogenous IPs, also increased apoptosis. Because IP6 prevents ectopic calcification, Addison and McKee first tested the effect of sodium IP6 on bone matrix mineralization in MC3T3-E1 cells grown under osteogenic conditions including AA and ß-glycerophosphate (ßGP) [42]. They found that sodium IP6 at concentrations of 3 μM or higher caused a marked reduction in ECM mineralization in association with upregulated SPP1, a potent inhibitor of mineralization. Notable was that neither osteoblast differentiation, alkaline phosphatase (ALP) activity, nor expression of the osteoblast marker genes Bglap and Ibsp was affected by sodium IP6 in this study.

Several follow-up studies have been conducted on the effects of IP6 supplementation in osteoblastic cell cultures. Treatment with IP6 (0.1 to 100 µM, Azopharma product) elicited different effects on osteoblastogenesis in MC3T3-E1 cells versus human mesenchymal stem cells (MSCs) [43]. Expression of the osteoblast marker genes Alpl, Ibsp, and Tnfsf11 decreased in the former cell model. Although both cell types were cultured under the same osteogenic conditions (AA, ßGP, and hydrocortisone), ALPL levels increased in the latter model. The authors suggested that this probably reflects differences in the ability of the different cell models to convert IP6 to pyrophosphate derivatives. When human MSCs were cultured in osteogenic medium including AA, ßGP, and dexamethasone (DEX), both ALP activity and amount of mineralized ECM increased in the presence of sodium IP6 at 5 and 10 µg/mL [44]. Human bone marrow MSCs (BMSCs) obtained from five patients with type 2 diabetes mellitus (DM) (aged 18–22 years) were cultured in the presence or absence of AA, ßGP, and DEX under conditions mimicking DM (supplementation with glucose and palmitic acid) [45]. In this cell culture model, calcium IP6, most effectively at a concentration of 34 μM, promoted not only cell proliferation, but also expression of ALPL, SPP1, RUNX2, and COL1, ALP activity, and ECM mineralization, possibly via the MAPK/JNK signaling pathway. Similarly, calcium IP6 (1%) ameliorated high-glucose-induced decreases in ALP activity, osteoblast marker gene expression, and ECM mineralization in human BMSC cultures in the presence of AA, ßGP, and DEX [46]. In addition, calcium IP6 prevented senescence of human BMSCs under high-glucose conditions, as evidenced by the repression of production of reactive oxygen species and senescence-associated ß-galactosidase, as well as a decrease in P21 and P53 expression. The authors then concluded that the ERK pathway may be involved in the effects of IP6. This group further pursued the mechanism underlying the rescue effects of IP6 on high glucose-mediated osteogenic dysfunction in human BMSC cultures (see [45,46]) and showed the involvement of the circEIF4B/miR-186-5p/FOXO1 and circEIF4B/IGF2BP3/ITGA5 axes [47]. In cultures of human periodontal ligament fibroblasts grown under osteogenic conditions (AA plus ßGP) with high glucose, IP6 at a pharmaceutical concentration (0.02%) increased calcium accumulation in the ECM, but decreased cell viability and ALP activity [48]. Thus, the prevailing view is that IP6 positively regulates osteoblasts and/or their precursor cells through certain signaling pathways at least under DM conditions, which seems to contradict the inhibitory effect of IP6 on ECM mineralization under osteogenic conditions. When evaluating the effects of IP6, it is necessary to separate non-cellular and cellular effects, and also to consider the differences in the effects of IP6 under physiological and pathological conditions.

To control implant osseointegration and survival, the chemical and biological properties of IP6-coated substrates have also attracted attention, and several studies have been conducted on the development of IP6 composite materials. Titanium directly coated with IP6 increased osteogenic activity of MC3T3-E1 cells and reduced the adhesion of S. mutans and S. sanguinis compared to titanium alone [49]. Similarly, human BMSCs seeded on IP6-magnesium multilayers [50], micro/nanostructured calcium IP6 [51], and calcium IP6 [52] bonded to titanium exhibited improved cell proliferation as well as increased expression of osteoblast marker genes, ALP activity, and calcium deposition under osteogenic conditions. The second composite also showed antibacterial properties against P. gingivalis, and the third reduced superoxide dismutase production and oxidative stress damage. Mouse BMSCs cultured on IP6/copper-coated titanium exhibited increased cell proliferation, expression of osteoblast marker genes, collagen secretion, and HA deposition, along with antibacterial activity against P. gingivalis [53]. IP6 and curcumin introduced into a wood-derived hydrogel composite membrane (artificial periosteum) elicited synergistic effects on cell adhesion, proliferation, ALP activity, and osteoblast marker gene expression in mouse BMSC cultures [54]. This artificial periosteum also exhibited antibacterial activity against E. coli and S. aureus, and inhibited and promoted the production of M1- and M2-phenotype factors, respectively, in the lipopolysaccharide-stimulated mouse macrophage cell line RAW264.7. Rat BMSCs were cultured on polyetheretherketone fibers (PKFs, as artificial ligaments) coated with IP6 or IP6/magnesium complex [55]. IP6 and IP6/magnesium bonded to PKFs enhanced cell adhesion, proliferation, osteoblast marker gene expression, ALP activity, and ECM mineralization, with the latter showing stronger effects. The authors suggest that the osteogenic effects seen in this model may be caused by IP6 and magnesium ions released from PKFs. In Chitin whisker/chitosan liquid crystal hydrogel-supported 3D-printed poly-L-lactide scaffolds soaked in IP6 solution (10 wt%) (PP-CHWs/CS-P), IP6 contributed to enhanced cell adhesion, ALP activity, and calcium nodule formation in BMSC cultures by increasing the modulus of the hydrogel matrix, and conferred bacteriostatic activities against E. coli and S. aureus via its antibacterial effect [56]. These findings suggest the utility of IP6 with hybrid materials for dental implants and bone regeneration.

In contrast, no positive effects of IP6 have been reported in other studies, i.e., in MC3T3-E1 cells with 500 µg/mL or more of 1% IP6 [57] and in rat bone marrow stromal cells with SNF472 at a maximum concentration of 30 μM [32]. In the former studies, no effects of IP6 were seen on cell proliferation and ALP activity in MC3T3-E1cells under non-osteogenic culture conditions. SNF472 elicited no changes in ALP and von Kossa staining, while these results may include non-osteoblastic calcification (the von Kossa-positive areas appear to be much more abundant than ALP-positive areas unexpectedly) in rat bone marrow stromal cells grown in the presence of AA and DEX. SNF472 showed biphasic effects on expression of Tnfrsf11b and Bcl2 genes, in contrast to no changes in the expression of Alpl, Bglap, Sost and Bax genes. Also, slight changes in Spp1 (decreased at low concentrations) and Sp7 (increased at high concentrations) were observed. It is acknowledged that it is not easy to mimic osteoblastogenesis and ECM mineralization in cell cultures of heterogeneous cell populations, or even those of osteoblasts and their progenitor cells without osteogenic stimuli. Treatment of the human osteosarcoma cell line SaOS-2 with ethanol extract (50 or 100 μg/mL) of brown rice (BRE) digested with Lactobacillus sakei Wikim001 (WK) increased cell viability and ALP activity compared to BRE not digested with WK [58]. Since WK has phytase activity and decreases IP6 in BRE, these results suggest that IP6 may adversely affect osteoblast development; however, no similar studies on normal cells have been conducted yet. 

**Table 1 biomolecules-14-01072-t001:** Studies on the effects of IP6 on bone cells in vitro.

Cell Type	IP6 Type	Effect		Model	Additional Information	Ref.
			Mode			
Osteoblasts	Sodium IP6	Nagative	Apoptosis	MC3T3 cells under serum starvation	IP6 at concentrations 10 times or higher than normal (50 µM) was used.	[41]
	Sodium IP6		Mineralization	MC3T3-E1 cells with osteogenic medium	Neither osteoblast differentiation, ALP activity, nor marker gene expression (except Spp1) was affected.	[42]
	Sodium IP6		Marker gene expression including Alpl	MC3T3-E1 cells with osteogenic medium	ALP activity was not affected.	[43]
	IP6	Positive	DNA synthesis	Newborn rat calvaria cells under mechanical stress	Intracellular IPs including IP6 were involved in the transduction of mechanical strain into cellular proliferation.	[37]
	Sodium IP6		ALPL gene expression	Human MSCs with osteogenic medium	−	[43]
	Sodium IP6		ALP activity, ECM mineralization	Human MSCs with osteogenic medium	−	[44]
	Ca IP6		Cell proliferation, marker gene expression, ALP activity, ECM mineralization	Human T2DM BMSCs with osteogenic medium/high glucose/palmitic acid	IP6 improved impaired osteoblastogenesis under culture conditions mimicking T2DM.	[45]
	Ca IP6		ALP activity, marker gene expression, ECM mineralization, cell senescence	Human MSCs with osteogenic medium/high glucose		[46]
	IP6		ECM mineralization (negative for cell viability, ALP activity)	Human PDL fibroblasts with osteogenic medium/high glucose		[48]
	IP6		ALP activity, marker gene expression	MC3T3-E1 cells on IP6-coated titanium	IP6 was introduced into titanium for implants and scaffolds for bone regeneration. It has been suggested that these effects may be mediated by the release of IP6 from carriers.	[49]
	IP6		Cell adhesion, cell proliferation, ALP activity, marker gene expression	Human BMSCs on IP6/Mg-coated titanium with or without osteogenic medium		[50]
	Ca IP6		ALP activity, ECM mineralization	Human BMSCs on micro/nanostructured Ca IP6-coated titanium with osteogenic medium		[51]
	Ca IP6		Cell proliferation, ALP activity, ECM mineralization, marker gene expression	Human BMSCs on Ca IP6-coated titanium with or without osteogenic medium		[52]
	IP6		Cell proliferation, marker gene expression, collagen secretion, ECM mineralization	Mouse BMSCs on IP6/copper-coated titanium		[53]
	IP6		Cell adhesion, cell proliferation, ALP activity, marker gene expression	Mouse BMSCs on IP6/curcumin-hydrogel composite membrane		[54]
	IP6		Cell adhesion, cell proliferation, marker gene expression, ALP activity, ECM mineralization	Rat BMSCs on IP6- and IP6/magnesium-coated PKFs		[55]
	IP6		Cell adhesion, ALP activity, ECM mineralization	BMSCs in IP6-soaked poly-L-lactide scaffolds		[56]
	SNF472	No effect	ALP activity, ECM mineralization	Rat bone marrow stromal cells with osteogenic medium	Bone marrow stromal cells include non-osteogenic cells. Some marker genes were affected.	[32]
	IP6		Cell viability, ALP activity	MC3T3-E1 cells	ALP activity decreased with days in culture, suggesting that this model does not mimic osteoblastogenesis.	[57]
Osteoclasts	IP6	Negative	Calcium release (bone resorption)	Organ cultures of rat long bones with PTH	−	[59]
	IP6		Osteoclast formation	RAW264.7 cells and human PBMCs with TNFSF11 and TNFSF11/CSF1/DEX, respectively	−	[60]
	BRE		Osteoclast formation	Mouse BMMs with TNFSF11/CSF1	−	[58]
	Sodium IP6	Positive	Osteoclast formation	BMMs, from rats fed high sodium IP6 and low calcium diet, with TNFSF11/CSF1	BMMs were isolated from rats with renal dysfunction, bone loss, etc. IP6 was not added to the culture.	[7]

ALP, alkaline phosphatase; BMMs, bone marrow macrophages; BMSCs, bone marrow MSCs; Ca, calcium; DEX, dexamethasone; ECM, extracellular matrix; Mg, magnesium; MSCs, mesenchymal stem cells; PDL, periodontal ligament; PKFs, polyetheretherketone fibers; PTH, parathyroid hormone; T2DM, type 2 diabetes mellitus.

In 1984, organ cultures of fetal SD rat long bones (radii and ulnae) labeled with ^45^Ca showed that IP6 and its hydrolyzed samples (IP1–IP5) at 10^−4^–10^−7^ M inhibited the release of ^45^Ca from cultured bones in the presence of PTH [59], suggesting that IPs may inhibit PTH-induced bone resorption. After a gap of more than 30 years, this result was supported by the findings that IP6 inhibited TNFSF11- and TNFSF11/CSF1/DEX-dependent osteoclast formation in RAW264.7 cells and human peripheral blood mononuclear cells, respectively [60]. However, IP6 had a biphasic effect with a maximum concentration of 1 μM, and the authors speculated that this may be due to the ability of the cells to convert IP6 to other IPs [60]. Treatment of mouse bone marrow macrophages (BMMs) with BRE, regardless of WK digestion, inhibited TNSF11/CSF1-dependent osteoclast formation; the effect of the latter was larger than that of the former, suggesting the inhibitory effect of IP6 in BRE on osteoclastogenesis [58]. Kim et al. prepared BMMs from SD rats fed control or high-sodium IP6-supplemented low-calcium diet for 12 weeks and cultured them with TNFSF11 and CSF1 [7]. In this model, the number of osteoclasts was higher in cultures of BMMs from rats fed 3% and 5% sodium IP6 diets than those fed 1% sodium IP6 and control diets, while it would be difficult to determine whether the effect is due to metabolic, renal, or bone abnormalities or to IP6. In a cell-free system, the inhibitory effect of sodium IP6 (1 μM and 3 μM) on HA dissolution could be seen; the effect was stronger than that seen with etidronate and comparable to that seen with alendronate, both of which are established antiosteoporosis agents [61]. Thus, while the majority of studies in culture models support the view that IP6 may inhibit osteoclast formation and promote osteoblast development, some conflicting results have been reported under some conditions. Further studies are needed to control for the proliferative and differentiation status of the cells used, the synthetic and metabolic capacity of IP6, and the properties of the IP6 used, amongst important variables.

## 3. The Effects of IP6 on Bone in Laboratory Animals

The use of controlled conditions, including the nutrient composition of the chow used, in laboratory animals, is helping to advance understanding of the complexity of IP6 activities and the underlying mechanisms in the organism. The effects of IP6 on bone in laboratory animals are summarized in Table 2. Wistar rats (3-week-old) fed a pure flour diet containing large amounts of phytin (0.13%, 2.17 times more than other diets tested) for 60 days exhibited growth retardation, hypocalcemia, low plasma ALP and 25-hydroxyvitamin D_3_ levels, decreased mineral (calcium, phosphate, and magnesium) content in the femur, and smaller skeletons and rib-cage deformities; supplementation with calcium improved, at least in part, these aberrations [62]. Of note, the pure flour diet used had large deficiencies in its nutrient profile compared to others tested, including lower amounts of vitamin D (6 times), calcium (more than 9.5 times), magnesium (1.4 times), and phosphorus (4.7 times). Therefore, it is difficult to determine whether these results are due to the influence of phytin. 

Studies with male CH3 mice (5-week-old) fed the L-488F diet containing 1% chromium (III) oxide with or without 1% or 5% sodium IP6 for 2 weeks showed no clear evidence for the effect of sodium IP6 on body weight gain, nitrogen, calcium, magnesium, and phosphorus balance or these elements in bone, except for the low availability of magnesium in mice fed 5% sodium IP6 [63]. The authors suggested that 2 weeks may have been insufficient to assess the effects of IP6 on bone. Weaning male SD rats were fed AIN-76 diets with varying calcium content (0.2, 0.4 and 0.8%) for 8 weeks to evaluate the effect of sodium IP6 (added to the diet in a molar ratio of 19:1 to calcium) on the femur, tibia, and mandible [64]. In this study, there was a lack of consistency in the outcomes of IP6 treatment, although sodium IP6 decreased bone mineral content (BMC) and calcium levels only in the 0.8% calcium diet group. The authors noted that excess sodium IP6 in the 0.8% calcium diet group may cause hypernatremia and decrease bone formation in this model, but this has not yet been confirmed. Grases et al. evaluated the distribution of minerals (calcium, magnesium, zinc, iron, manganese) in bone, urine, and other tissues of 16-week-old Wistar rats fed the AIN-76A diet with or without sodium IP6 (1%) [65]. These rats were also either exposed to or not exposed to IP6 during fetal and lactation periods by feeding their parents with the same diet as above. The authors then concluded that dietary sodium IP6 only affected bone zinc levels, observing in particular that femur zinc levels in rats fed the AIN-76A diet were 10-fold higher than those in rats fed the common non-purified rat diet (containing 0.8% IP6). 

In young male SD rats fed the low-zinc AIN-93 diet including 0.3% sodium IP6 with or without phytase (1500 FTU/kg) for 8 weeks, dual-energy X-ray absorptiometry (DXA) revealed that phytase supplementation increased lean body mass, and femoral/tibial BMC and strength, in parallel with increased plasma and femur zinc levels [66]. Taken together with the lack of differences in iron, calcium, manganese, and magnesium status between the groups tested, the authors suggested that the positive effects of phytase supplementation may be the result of improved zinc nutriture. When combined with the results of MC3T3 cell [41], SaOS-2 cell [58], and bone organ cultures [59], it seems necessary to assess whether hydrolysates of IP6 by phytase (IP1–IP5) are also involved in bone cell activities. In Wistar rats (average body weights, 236.4 g) fed the AIN-93G diet including AIN-93G mineral/vitamin mix and with or without zinc carbonate for 3 weeks, supplementation with IP6 extracted from sweet potato (Ipomoea batatas) or commercially available IP6 (the ratio of IP6 to zinc, 18:1) yielded inconsistent results in fecal and femur calcium, iron, magnesium, and zinc levels [67]. However, scanning electron microscopy revealed that IP6, whether extract or commercial, reduced trabecular bone thickness in the femur. The reasons for this discrepancy are unclear, but microcomputed tomography (μCT) and bone histomorphometry are recommended to evaluate bone parameters. In the report of Kim et al. [7], SD rats fed the AIN-93G diet with 3% or 5% sodium IP6 but not 1% sodium IP6 exhibited bone anomalies, such as decreased bone mineral density (BMD) (by DXA), decreased bone parameters (by μCT and bone histomorphometry), and increased numbers of osteoclasts in the femur. The authors reported that increased phosphorus levels by the hydrolysis of IP6 in the intestine were involved in these anomalies which were improved by calcium supplementation. When SNF472 (25 mg/kg) was infused intravenously (15 min/dose) three times a week for 9 months in Beagle dogs, and bone histomorphometry of femoral cortical and trabecular bone was performed, no differences were seen in bone parameters or osteoclast numbers between the SNF472-treated and control groups [32]. However, as the authors noted, SNF472 rapidly binds to HA and inhibits its crystal growth, making it important to measure mineralization parameters such as mineral apposition rate by, e.g., calcein labeling. Taken together, the results point to the need for further studies into the nutritional or health conditions under which excessive IP6 intake increases the risk of bone loss.

**Table 2 biomolecules-14-01072-t002:** Studies on the effects of IP6 on the bone in laboratory animals.

IP6			Effect		Model			Additional Information	Ref.
Type	Dose	Ad Route	Mode		Animal	Diet	Period		
Phytin	0.13%	Oral	Negative	Mineral content, skeleton size, rib-cage deformities	Wistar rats (3-wk-old)	Pure flour	60 d	The pure flour diet has large gaps in its nutrient profile, including low levels of Ca and vitamin D, compared to control diets tested.	[62]
IP6	The ratio of IP6 to Zn, 18:1	Oral		Tr Thk	Adult Wistar rats	AIN-93G + vitamin/mineral mix, ±Zn	3 wk	Ca and Mg contents were unaffected. Tr ThK was measured by scanning electron microscopy.	[67]
Sodium IP6	1, 3 or 5%	Oral		BMD, BV/TV, BS/TV, Tr No, osteoclast numbers, TNFSF11, TNFRSF11B, etc.	Weaning SD rats	AIN-93G	12 wk	IP6 (3% and 5%) may represent more than 4 times the average human intake in the US and UK. Ca supplementation improved skeletal anomalies in 3% IP6. No effect was seen in 1% IP6.	[7]
Ca/Mg IP6	12.99 g/kg	Oral	Positive	Ca and P contents, BMD, Urinary DPD (decreased)	OVX Wistar rats	AIN-76A	12 wk	These have been evaluated in models of diseases and bone defects.	[68]
Ca IP6	Dental repair films with 10 μL of 34 μM Ca IP6	Scaffold		BV/TV	Parietal bone defects in T2DM-SD rats	HFD + STZ	8 wk		[45]
IP6	Human BMSCs with 4 μM IP6 and collagens	Local		BV/TV, BS/TV, Tr No, Tr Thk, MAR, osteoblast markers.	Cranial defects in T2DM-male SD rats	HFD + STZ	12 wk		[47]
IP6	Mouse BMSCs with 4 μM IP6	Local		BV/TV	Tooth extraction sites in T2DM-male C57/BL6 mice	HFD + STZ	9 d		[46]
IP6	IP6- or IP6/Mg-coated PKFs with rat BMSCs	Scaffold		Accumulation of M2 macrophages involved in bone formation, BMD and BV/TV	Air pouches in SD rats Ligament-bone defects in New Zealand rabbits	−	14 d 12 wk		[55]
IP6	IP6/Calcium-decorated titanium	Scaffold		BV, Tr No, Tr Sep, bone implant contact	Bone marrow cavities in SD rats	HFD + STZ	8 wk		[52]
IP6	IP6-soaked poly-L-lactide	Scaffold		Osteoid and mineralized bone areas	Muscle pockets in male SD rats	−	6 wk		[56]
Sodium IP6	3, or 5%	Oral	No effect	Bone ash, Ca, Mg, and P contents	Male CH3 mice (5-wk-old)	L-488F + 1% Cr_2_O_3_	2 wk	−	[63]
Sodium IP6	0.15, 0.3, or 0.6%	Oral		BMC, Ca and P contents, bone histomorphometric parameters	Weaning male SD rats	AIN-76 + 0.2, 0.4, or 0.8% Ca	8 wk	Negative effects were seen only in 0.8% Ca and 0.6% sodium IP6. The authors did not clarify whether this was due to IP6 or sodium.	[64]
Sodium IP6	1%	Oral		Ca, Mg, Fe, and Mn contents	Wistar rats	AIN-76A	16 wk	Zn content was affected. Zn levels in this model were 10-fold higher than those in rats fed the common non-purified diet.	[65]
SNF472	25 mg/kg, 3 times weekly	IV infusion		BV, Tr No, Tr Thk, cortical Thk, osteoclasts, etc.	Beagle dogs	−	9 mth	−	[66]

Ad, administration; BMC, bone mineral content; BMD, bone mineral density; BMSCs, bone marrow mesenchymal stem cells; BS, bone surface; BV, bone volume; Ca, calcium; d, days; DPD, deoxypyridinoline; Fe, iron; HFD, high fat diet; IV, intravenous; MAR, mineral apposition rate; Mg, magnesium; Mn, manganese; mth, months; OVX, ovariectomized; P, phosphorus; T2DM, type 2 diabetes mellitus; Tr No, trabecular number; Tr Sep, trabecular separation; Tr Thk, trabecular thickness; TV, tissue volume; PKFs, polyetheretherketone fibers; STZ, streptozotocin; wk, weeks; Zn, zinc.

Ovariectomized Wistar rats, a postmenopausal osteoporosis model, were fed the AIN-76A diet including or excluding 12.99 g/kg calcium/magnesium IP6 for 12 weeks [68]. DXA showed that rats fed calcium/magnesium IP6 had greater BMD in the femur, concomitant with increased levels of calcium and phosphorus in bones (L4 vertebra and femur) and decreased levels of urinary deoxypyridinoline, a bone resorption marker, compared to those without calcium/magnesium IP6. Lv et al. evaluated the effect of calcium IP6 on surgically created bone defects in a type 2 DM rat model in which SD rats were fed a high-fat diet for 4 weeks, and then injected intraperitoneally with streptozotocin, leading to high blood glucose levels [45]. In these rats, dental repair films with or without calcium IP6 (10 μL of 34 μM) were placed in parietal bone defect sites; healing of the bone defect was accelerated in the IP6-treated group compared to the non-treated group. Using μCT and histological examinations in a type 2 DM rat model, Wu et al. provided evidence that transplantation of collagen mixed with human BMSCs (10^7^ cells) cultured for 21 days in the presence of 4 μM IP6 into cranial defects, compared to that in the absence of IP6, accelerated bone regeneration and that circEIF4B silencing by sh-circEIF4b in human BMSC cultures with IP6 (see Section 2) attenuated the effectiveness of IP6 treatment [47]. The in vivo effects of IP6 were also demonstrated in a type 2 DM mouse model (C57BL/6) [46], generated as above [45,47]. Three injections of mouse BMSCs cultured with 4 μM IP6 for 7 days (10^6^ cells) into the extraction sockets of the maxillary first molars, compared to control BMSCs, promoted de novo bone formation in the sockets. These findings indicate that IP6 may improve bone formation and bone turnover under certain pathological conditions. A certain number of negative effects of IP6 on bone have been reported in both laboratory animal and in vitro experiments, but there is currently no evidence that these effects are related to animal species and strains, cell types, IP6 types, or other specific conditions. In any case, it is necessary to further brush up the experimental conditions, such as examining the correlation between blood concentrations of IP6 and its effects on bone and the impact of reduced IP6 intake on bone, in order to more rigorously evaluate IP6 intake appropriate for the human dietary environment.

IP6 composite scaffolds have been evaluated for their effects on bone formation in vivo. μCT demonstrated that the transplantation of PKFs with IP6 into anterior cruciate ligament and femoral/tibial defects in New Zealand rabbits (1.5–2 kg) increased newly formed bone volume/tissue volume and BMD vs. that of PKFs without IP6 [55]. In this model, the authors also showed that the addition of IP6 suppressed the formation of fibrous tissue by PKFs and contributed to the accumulation of M2 macrophages involved in bone formation rather than M1 macrophages responsible for inflammation and that these effects of IP6 were enhanced in combination with magnesium. These results suggest that IP6 may contribute to ligament–bone healing by targeting multiple sites. Similar results were obtained after implantation of calcium/IP6-decorated titanium into the bone marrow cavity through the distal end of the femur in a rat model of type 2 DM [52]. PP-CHWs/CS-P scaffolds with or without IP6 were implanted into the thigh muscle pockets of male SD rats (7-week-old), and histological and immunohistochemical examinations were performed to evaluate their osteogenic capacity [56]. The results of this study showed that the introduction of IP6 into the scaffolds increased the number of the osteoblast marker COL1-positive cells and expanded the osteoid and mineralized areas. Thus, these approaches may represent one direction for the effective use of IP6, independent of the nutritional arguments regarding excess IP6 intake.

## 4. The Effects of IP6 on Bone in Humans

In humans, IP6 has been evaluated primarily on the basis of normal dietary intake or urinary IP6 concentrations in healthy subjects or osteoporosis patients, and so far there have been no reports of negative effects of IP6 on bone within this range. The majority of excellent studies came from the group of Prof. Felix Grases, University of Balearic Island, Mallorca, Spain, based on smart and elegant studies. The BMD of the calcaneus of 1473 Spanish (Mallorca) volunteer workers belonging to different activity sections (18–65-year-old) and of the lumbar column/femoral neck of 433 subjects randomly selected from the cohort was evaluated using DXA. Notably, the BMD was higher depending on the estimated dietary intake of IP6 (based on the sum of servings consumed per week of the main food groups—legumes, nuts, and whole-grain food including whole-meal bread—that contain IP6) [69]. The subanalysis of the above study showed similar results on the relationship between frequency of phytin consumption per week and calcaneus BMD in 433 postmenopausal women (203 of these subjects also had L2–L4 lumbar spine and femoral neck BMD) [70]. A prospective study also demonstrated a correlation between estimated dietary intake of IP6 from the Mediterranean diet and lumbar spine (L1–L4) BMD in postmenopausal women in Palma de Mallorca, Spain [61]. In this study, BMD in women who take 200–325 mg/day IP6 (average age 55.4, 127 subjects) and over 325 mg/day IP6 (average age 55, 147 subjects) was higher than that in those consuming less than 199 mg/day IP6 (average age 56.6, 145 subjects). Thus, the authors postulated that IP6 may suppress bone resorption, because sodium IP6 (1 or 3 μM) inhibited HA dissolution under acidic conditions at levels comparable to that seen with alendronate. A similar finding on the relationship between estimated dietary intake of IP6 and femoral and lumbar spine bone parameters was obtained in a study of 561 postmenopausal women [71]. These subjects with data on bone parameters in the femur and lumbar spine at baseline were a subset of a larger cohort of 6874 males and females aged 55–75 years who were overweight and obese (body mass index 27 kg/m^2^ to 40 kg/m^2^) and met at least three features of the metabolic syndrome (raised blood pressure, dyslipidemia, raised fasting glucose, central obesity, etc.) in a 6-year, randomized, parallel-group, multicenter, controlled study on primary prevention of cardiovascular disease that is ongoing in Spain (the Prevención con Dieta Mediterránea Plus, Spain). 

The correlation between IP6 and BMD was more rigorously evaluated, i.e., the relationship with BMD was evaluated using urinary IP6 levels, rather than estimated levels of IP6 intake. Among 180 postmenopausal volunteers in Mallorca, 140 subjects with low levels of urinary IP6 (<0.88 mg/L, average age 52.4) and 40 subjects with high levels of urinary IP6 (>1.33 mg/L, average age 51.6) underwent DXA scanning to measure their lumbar spine (L2–L4) and femoral neck BMD [21]. It was found that BMD, together with body weight, in the group with high IP6 levels was higher than that seen in the group with low IP6 levels. A similar result was found in the correlation of urinary IP6 levels with L2–L4 and femoral neck BMD and further with 10-year fracture risk, using DXA and the tool FRAX^®^, respectively, in 94 Spanish postmenopausal women (Mallorca) [22]. Of 143 postmenopausal women (Balearic Islands, Majorca, Spain), femoral neck BMD in 52 women with high urinary IP6 levels (≥1.0 mg/L, average age mean 51) was higher than that in 91 women with low urinary IP6 levels (≤0.5 mg/L, average age 52) [23]. In addition, the fracture risk assessment tool model showed that the former had a lower 10-year fracture risk than the latter, when any one of the osteoporosis risk factors (e.g., tobacco, alcohol, and well-food intake) were present. However, it remains unclear whether this is due to the inhibitory effect on bone resorption, a stimulatory effect of bone formation, or a direct or indirect effect on bone. 

In a review of the safety of soy-based infant formulas (SIFs) in children, anthropometric growth, bone health (bone mineral content), immunity, cognition, and reproductive and endocrine functions were described in children fed SIFs versus those fed other types of infant formulas (soy IP6 exists as the potassium salt) [72]. The authors concluded that there were no differences in any of the parameters studied between the children fed SIFs versus other infant formulas. The effect of SNF472, a novel calcification inhibitor to treat vascular calcification and calciphylaxis, on the BMD in male and female dialysis patients with coronary artery calcification has been reported for doses of 300 mg (75 subjects/30 female, average age 63) and 600 mg (60 subjects/23 female, average age 64) of SNF472 or placebo (67 subjects/22 female, average age 64) administered intravenously three times a week for 52 weeks [30]. Serum ALP, calcium, magnesium, phosphate, and intact PTH did not differ among the groups, while DXA revealed decreased proximal femoral BMD in the SNF472 600 mg group. Fractures were rarely reported in either group. The authors stated that this study limited the ability to determine the clinical significance of the changes observed, because changes in BMD and rates of fracture are generally assessed over multiyear studies and discuss the need for long-term monitoring of bone, including bone quality. Of note, mineral bone disease is a common complication of chronic kidney disease characterized by defects in mineral homeostasis, including hypocalcemia, hyperphosphatemia, decreased vitamin D, and bone loss [73], but there is currently no clear evidence that SNF472 exacerbates these defects in dialysis patients. A small group of patients with osteopenia or osteoporosis, calciuria, and with increased bone resorption and kidney stones was treated with calcium/magnesium IP6 (Broken^®^ Salvat Laboratories) as a dietary supplement [74]. Of 23 patients in this study (18–65 years old, female and male in the Son Espases University Hospital, Mallorca), 12 (male, 7) took 380 mg/day of IP6 for 3 months and 11 (male, 6) did not (control). These patients received no other treatments with bisphosphonates, thiazides, citrates, or glucocorticoids, had no access to calcium and/or vitamin D supplements and limited dietary intakes of IP6. In the IP6 intake group, calciuria and levels of the bone resorption marker ß-crosslaps (CTX) were improved. The authors noted that this was a short study with a small number of subjects and that dietary consumption of IP6 before and during the intervention was not quantified. The consumption of IP6 could modify the results. 

## 5. Conclusions and Perspectives

Bone development and homeostasis are influenced by a wide variety of nutrients, e.g., calcium, potassium, magnesium, phosphorus, zinc, manganese, copper, fluoride, sodium, protein, vitamins, caffeine, fiber, omega-3 fatty acids, and alkali-rich foods [75,76]. We focused on the direct effects of IP6 on bone, but the effects of IP6 on these nutrients should also be considered. See, for example, two excellent reviews on the effects of IP6 on mineral bioavailability; some studies have shown that mineral absorption was not altered by IP6, while others have shown IP6-dependent decreases in mineral bioavailability [12,77]. As discussed by Kunkel et al. [64], if excessive intake of sodium IP6 induces hypernatremia, it leads to bone loss [78]. So far, there has not been solid evidence for this effect in humans. However, as noted above, excess sodium may need to be considered carefully when using IP6 sodium; findings associated with hypernatremia such as hypocalcemia, hypophosphatemia, elevated serum PTH levels, and increased number of osteoclasts with bone loss have been observed in rats fed excessive amounts of sodium IP6 [7]. In vitro, on the other hand, sodium concentrations in the medium does not change significantly when sodium IP6 is given even at a concentration of 100 µM, suggesting that the results in a cell culture model reflect IP6 actions. Further attention should be paid to this issue in the future. 

Currently, there are limited studies showing that IP6 directly impairs bone development and/or homeostasis in humans or laboratory animals; rather, there are many studies on the beneficial effects of IP6 on bones. Therefore, except under certain conditions (e.g., malnutrition), the positive effects of IP6 on bones are expected to outweigh the negative effects (Figure 1). IP6 binds to HA via its negatively charged phosphate groups and inhibits HA crystal growth. This action proceeds extracellularly and can be mimicked in a model of ECM calcification in osteoblast cultures. When IP6 and/or its metabolites [79] are produced in cells or incorporated into cells, these molecules may impinge on various signaling pathways with consequent modulations of biological activities. Thus, it is plausible that IP6 and/or its metabolites may act on various intracellular and extracellular target molecules involved in bone development and homeostasis. It remains poorly understood how the intracellular and extracellular actions of IP6 contribute to these outcomes. It is also unclear whether the difference in counterions (salts) of IP6 affects the direct actions of IP6 on bone, although plant-derived calcium/magnesium IP6 and sodium IP6 show differences in calcium bioavailability [5]. 

Taking advantage of what is known about IP6 (good safety, tolerability, and encouraging efficacy in clinical studies), the development of IP6 derivatives for clinical use is underway with focus currently on improving pathological calcification [80]. In materials science, organic modification of IP6 was carried out by chemical anchorage of glyceryl motility to reduce the number of ionized protons that modulate chelating activity [81,82]. This modification showed good biological properties in osteoblast cultures. We expect that the accumulation of data from basic studies and large-scale clinical trials with standardized dosing regimens will help to unravel the inconsistencies and unknowns regarding the effects of IP6 on osteoblasts and osteoclasts as well as bone metabolism in humans, such that the use of IP6 and its derivatives may be effectively used to increase bone mass and improve bone quality.

## Figures and Tables

**Figure 1 biomolecules-14-01072-f001:**
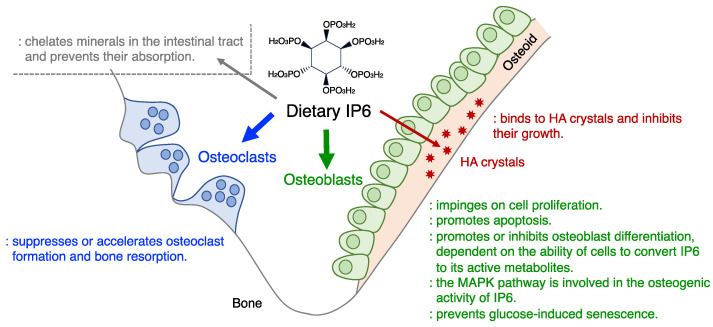
The possible roles of dietary IP6 in bone. Descriptions are primarily based on in vitro evaluations. The size of the arrows indicates the magnitude of the effect of IP6 as inferred from in vivo studies including those in humans. So far, studies with dietary IP6 report primarily beneficial effects on bone health and disease, but some contradictory test results should not be ignored.

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
