# Peer review of "Inositol Hexaphosphate in Bone Health and Disease"

_biomolecules, 2024, doi:10.3390/biom14091072_

Round 1

Reviewer 1 Report (Previous Reviewer 2)

Comments and Suggestions for Authors

The authors have adequately addressed the comments and suggestions.

Author Response

We appreciate the reviewer's comment.

Reviewer 2 Report (Previous Reviewer 1)

Comments and Suggestions for Authors

To the Editor

The reviewed manuscript brings improvements to the review about the role of dietary phytic acid (myo-inositol hexaphosphate, IP6) in bone health and disease. It explores IP6’s historical perspective as an antinutrient due to its mineral chelating activity, and its emerging beneficial effects on various biological processes, including anti-inflammatory, antibacterial, and anticancer properties. The paper reviews the evidence suggesting IP6’s involvement in bone health, focusing on its interaction with hydroxyapatite crystals and its implications for both pathological and physiological mineralization.

Despite the improvements, the manuscript is still too weak to be considered a review. Please, see below my major and minor concerns

Major Comments

1. Depth of Literature Review: The literature review is thorough, covering historical and recent studies on IP6’s impact on bone health. However, the manuscript could benefit from a more structured approach to summarizing conflicting findings and providing a clearer rationale for the observed discrepancies. 

1.1. Organize the literature review into clear sections that address different aspects of IP6’s impact on bone health. Consider using subheadings for in vitro studies, animal studies, and human studies.

2. Methodology: The review primarily synthesizes findings from various in vitro, animal, and human studies. While this approach is appropriate for a review, a more critical evaluation of the methodologies used in the cited studies would enhance the manuscript. For instance, discussing the differences in IP6 forms (sodium vs. calcium/magnesium salts) and their implications on the study outcomes would be beneficial.

2.1. Provide a critical evaluation of the methodologies used in the cited studies, highlighting the strengths and weaknesses of different approaches.

3. Discussion and Interpretation: The discussion effectively highlights the potential mechanisms through which IP6 influences bone health. However, the manuscript should address the limitations of the current evidence more explicitly, including the variability in study designs, doses of IP6 used, and the potential for publication bias.

3.1. Discuss the limitations of the current evidence more explicitly, including potential sources of bias and the variability in study designs.

4. Conclusion: The conclusion succinctly summarizes the potential benefits of IP6 for bone health, emphasizing the need for further research. It would be helpful to provide specific recommendations for future studies, such as standardized dosing regimens and more extensive clinical trials to validate the findings.

Minor Comments

1. Abstract: The abstract is well-written but could be more concise. Including specific findings and their implications in the abstract would provide a clearer summary of the manuscript.

2. Figures and Tables: The manuscript includes a figure summarizing the roles of IP6 in bone health. Additional figures or tables comparing the outcomes of key studies would enhance the reader’s understanding of the evidence landscape. Consider adding more visual aids, such as tables summarizing key study outcomes or additional figures illustrating the proposed mechanisms of IP6 action.

3. Writing and Formatting: The manuscript is generally well-written, but minor grammatical errors and typographical issues should be addressed. For instance, “susuggest” should be corrected to “suggest” in line 136.

Decision

Based on the strengths and weaknesses outlined above, I recommend the authors address the major/and minor comments regarding the structure of the literature review, methodological critique, and detailed discussion of limitations will significantly enhance the manuscript’s quality and impact.

Therefore, for now, I REJECT this manuscript and will be happy to reconsider my decision upon extensive major reviews.

Comments on the Quality of English Language Minor editing of English language required

Author Response

We thank for the reviewer’s comments and suggestions. We have responded to each of the comments by making the following revisions. We hope that these revisions will improve the value of this paper and that our paper will be published in biomplecules.

  1. Depth of Literature Review: The literature review is thorough, covering historical and recent studies on IP6’s impact on bone health. However, the manuscript could benefit from a more structured approach to summarizing conflicting findings and providing a clearer rationale for the observed discrepancies.

1.1. Organize the literature review into clear sections that address different aspects of IP6’s impact on bone health. Consider using subheadings for in vitro studies, animal studies, and human studies.

Response

Since we believe that it is not necessarily appropriate to compare current more advanced experiments equally with those of the past, each item is listed in the order of the papers with the oldest year of publication. However, as pointed out by the reviewer, conflicting findings are summarized in separate paragraphs. In addition, paragraphs are divided by differences in target cells, approaches, etc., and issues are introduced in the first sentence. Therefore, no subheadings have been given. Instead, in the in vitro and experimental animal sections, where the descriptions are relatively complex, tables were added to facilitate comparison.

  1. Methodology: The review primarily synthesizes findings from various in vitro, animal, and human studies. While this approach is appropriate for a review, a more critical evaluation of the methodologies used in the cited studies would enhance the manuscript. For instance, discussing the differences in IP6 forms (sodium vs. calcium/magnesium salts) and their implications on the study outcomes would be beneficial.

2.1. Provide a critical evaluation of the methodologies used in the cited studies, highlighting the strengths and weaknesses of different approaches.

Response

As the reviewer noted, we have also mentioned the methodology, but we have now added more (please see P5 line 186~; P6 line 219~, line 223~, and line 254~; P7 line 276~, line 281~, and line 310~; P9 line 348~; P10 line 370~; P11 line 454~). As an example, the reviewer notes that it would be useful to discuss the different forms of IP6 (sodium salt vs. calcium/magnesium salt) and their impact on test results. We could not find enough number of papers to mention this issue.

  1. Discussion and Interpretation: The discussion effectively highlights the potential mechanisms through which IP6 influences bone health. However, the manuscript should address the limitations of the current evidence more explicitly, including the variability in study designs, doses of IP6 used, and the potential for publication bias.

3.1. Discuss the limitations of the current evidence more explicitly, including potential sources of bias and the variability in study designs.

Response

Based on the reviewer's comments, the tables have been prepared according to the descriptions in the revised text (please see our response to 2. Methodology). In particular, the additional information indicates the issues involved in the experiments. We believe that these tables have clarified the contents of this paper.

  1. Conclusion: The conclusion succinctly summarizes the potential benefits of IP6 for bone health, emphasizing the need for further research. It would be helpful to provide specific recommendations for future studies, such as standardized dosing regimens and more extensive clinical trials to validate the findings.

Response

We have revised the descriptions, according to the reviewer's suggestions.

Minor Comments

  1. Figures and Tables: The manuscript includes a figure summarizing the roles of IP6 in bone health. Additional figures or tables comparing the outcomes of key studies would enhance the reader’s understanding of the evidence landscape. Consider adding more visual aids, such as tables summarizing key study outcomes or additional figures illustrating the proposed mechanisms of IP6 action

Response

We thank the reviewers for their suggestions. As noted above, we have added two tables and moved Figure 1 to the CONCLUSION section.

  1. Writing and Formatting: The manuscript is generally well-written, but minor grammatical errors and typographical issues should be addressed. For instance, “susuggest” should be corrected to “suggest” in line 136.

Response

I apologize for the multiple typos. The revised version has been proofread.

Reviewer 3 Report (New Reviewer)

Comments and Suggestions for Authors

Comments on the Quality of English Language

Extensive editing is needed.

Author Response

We thank for the reviewer’s comments and suggestions. We have responded to each of the comments by making the following revisions. We hope that these revisions will improve the value of this paper and that our paper will be published in biomplecules.

Round 2

Reviewer 2 Report (Previous Reviewer 1)

Comments and Suggestions for Authors

Dear Editor,

As I can see, the authors made substantial changes to improve their manuscript's quality and clarity (it does not mean new improvements could not be provided before the final form). Therefore, I recommend this manuscript for publication.

Best,

Wender

Author Response

We thank the reviewer for hisorher helpful comments and suggestions.

Reviewer 3 Report (New Reviewer)

Comments and Suggestions for Authors

Comments on the revised Manuscript:

1. The typographical errors need to be corrected as requested, e.g., lines 37, 38, 69, 75, 137, 242, 282, 441, 469, 509, etc. The authors should carefully check the entire text and make all the corrections.

Delete the following (line 241): “IP6 act on bone resorption in vitro.”

Comments on the Quality of English Language

N/A

Author Response

We thank the reviewer for his/her detailed comments. We apologize for the many typographical errors. We have corrected them using a spell checker.

This manuscript is a resubmission of an earlier submission. The following is a list of the peer review reports and author responses from that submission.

Round 1

Reviewer 1 Report

Comments and Suggestions for Authors

This paper is not a comprehensive review of the effects of IP6 (phytic acid/phytate/myo-inositol hexaphosphate) on bone metabolism in mammals. In my opinion, it sounds like a short review including both laboratory animals and humans. While the content of the paper seems relevant and informative, there are several reasons why it might not be suitable for publication in its current form:

A comprehensive review MUST be provided by the authors (last 20 years) since its current form is "poor" of proper citations: The paper lacks proper citations throughout the text, which is crucial for providing evidence and credibility to the claims and findings.

While the paper discusses various studies and findings related to IP6 and its effects on bone metabolism, it lacks a thorough discussion of the limitations of these studies and the potential biases or confounding factors that might affect their results.

Unfocused Conclusion: The conclusion lacks a clear summary of the key findings and their implications. It should also include suggestions for future research directions based on the gaps identified in the existing literature.

Overall, while the content of the paper appears informative, it requires significant revisions to meet the standards expected for a review article, including proper citation of sources, improved structure, and a more focused and concise presentation of information.

Thus, at this moment, I suggest rejecting this manuscript and further resubmission after extensive reviews.

Comments on the Quality of English Language

Moderate editing of English language required

Reviewer 2 Report

Comments and Suggestions for Authors

This paper presents a comprehensive review of the effects of inositol phosphates in bone health and disease. This is an interesting and current topic. Thus, for a long period of time, phytate (IP6) has been considered an antinutrient, responsible for the poor absorption of trace elements such as calcium, which can be associated with rickets.

a) The paper is well organized, although section 2 (Calcification of bone) should be integrated into the Introduction section. Then, the Review would include three fundamental sections:

-The effects of IP6 on bone in vitro.

-The effects of IP6 on bone in laboratory animals.

-The effects of IP6 on bone in humans.

The confusion that has been generated for a long time about the so-called antinutrient capacity of IP6 is a consequence of a series of errors related to the inositol phosphate salt used. Thus, most seeds contain phytate in the form of calcium/magnesium salt, except for soybeans, which contain it mainly as potassium salt. It is not found in nature as sodium phytate. Therefore, the consumption of IP6 in its natural form can hardly cause rickets, since it has an associated supply of calcium, and in fact nuts are recommended to strengthen bones. On the other hand, if high doses of IP6 are supplied in the form of sodium salt, apart from the effects of the phytate ion, the effects derived from high sodium consumption must be also considered. For this reason,

b) throughout the entire review it is necessary to specify the IP6 salt used, as well as its dose. For example, in section 4, line 21, the authors comment that in a study with Wistar rats fed IP6-enriched flour for 60 days exhibited growth retardation, hypokalemia, decreased mineral (calcium and phosphate) in the femur, etc. It would be necessary to indicate what amount of IP6 was added and in what salt form.

c) In the Conclusion section, the results of using the calcium-magnesium salt of IP6 (most common natural form) versus the sodium salt should be compared and discussed.

d) The results obtained with different IP6 sodium salt amounts should also be compared.

e) Possible effects due to very high sodium consumption should also be commented.

Specific comments:

1. Section 2 (calcification of bone) should be included in the introduction of the review, so it should be removed.

2.
The conclusions must be considerably modified and expanded, in accordance with the indications that have been commented to the authors of the review.